# Cutoff CT value can identify upper gastrointestinal bleeding on postmortem CT: Development and validation study

**Naomasa Okimoto**[1☯], **Masanori Ishida**[1☯], **Wataru Gonoi**[1☯]*, **Kotaro Fujimoto**[1],
**Keisuke Nyunoya**[1], **Mariko Kurokawa**[1], **Go Shirota**[1], **Hiroyuki Abe**[2], **Tetsuo Ushiku**[2],
**Osamu Abe**[1]

1 Department of Radiology, Graduate School of Medicine, The University of Tokyo, Hongo, Bunkyo-ku, Tokyo, Japan, 2 Department of Pathology, Graduate School of Medicine, The University of Tokyo, Hongo, Bunkyo-ku, Tokyo, Japan

☯ These authors contributed equally to this work.
* gonoiw@gmail.com

**Data Availability Statement:** We have uploaded the minimal anonymized data set as Supporting Information files in this submitting system.

## Abstract

This study aimed to establish the diagnostic criteria for upper gastrointestinal bleeding (UGIB) using postmortem computed tomography (PMCT). This case-control study enrolled 27 consecutive patients with autopsy-proven UGIB and 170 of the 566 patients without UGIB who died in a university hospital in Japan after treatment and underwent both noncontrast PMCT and conventional autopsy between 2009 and 2020. Patients were randomly allocated to two groups: derivation and validation sets. Imaging findings of the upper gastrointestinal contents, including CT values, were recorded and evaluated for their power to diagnose UGIB in the derivation set and validated in the validation set. In the derivation set, the mean CT value of the upper gastrointestinal contents was 48.2 Hounsfield units (HU) and 22.8 HU in cases with and without UGIB. The optimal cutoff CT value for diagnosing UGIB was ≥27.7 HU derived from the receiver operating characteristic curve analysis (sensitivity, 91.7%; specificity, 81.2%; area under the curve, 0.898). In the validation set, the sensitivity and specificity in diagnosing UGIB for the CT cutoff value of ≥27.7 HU were 84.6% and 77.6%, respectively. In addition to the CT value of ≥27.7 HU, PMCT findings of solid-natured gastrointestinal content and intra/peri-content bubbles ≥4 mm, extracted from the derivation set, increased the specificity for UGIB (96.5% and 98.8%, respectively) but decreased the sensitivity (61.5% and 38.5%, respectively) in the validation set. In diagnosing UGIB on noncontrast PMCT, the cutoff CT value of ≥27.7 HU and solid gastrointestinal content were valid and reproducible diagnostic criteria.

## Introduction

Computed tomography (CT) is a medical imaging technique that uses X-rays and computer processing to create detailed cross-sectional images of the body. Postmortem computed tomography (PMCT) is an important non-invasive method for examining the cause of death

**Funding:** This study was supported by Grants-in-Aid for Scientific Research (KAKENHI, JSPS; Grant numbers 20K07989 (received by WG) and 23K07202 (received by MI)). https://www.jsps.go.jp/j-grantsinaid/ The funder did not play a role in the study design, data collection and analysis, decision to publish, or preparation of the manuscript.

**Competing interests:** The authors have declared that no competing interests exist.

in the field of forensic medicine [1,2]. PMCT requires less cost and time, is more acceptable to bereaved families than conventional autopsies, and provides helpful information for subsequent conventional autopsies. Moreover, it is a substitute for conventional autopsies in certain situations [3–5].

Upper gastrointestinal bleeding (UGIB) is one of the most common disease entities, with a mortality rate of up to 10% [6]. Findings on noncontrast antemortem CT, such as clotted hematoma and elevated CT density of the gastrointestinal contents, are useful in identifying the gastrointestinal bleeding site [7–10]. However, distinguishing hematomas from other high-density gastrointestinal contents (e.g., food residues and medications) remains challenging [11].

As the endoscopic examination is not practical after death and autopsy is not performed in all cases of death, previous case reports indicated the capability of noncontrast-enhanced or contrast-enhanced PMCT to identify the gastrointestinal bleeding site [9,12,13]. The role of an increased CT density in gastrointestinal contents has been emphasized in UGIB diagnosis. Therefore, in the present case-control study, we aimed to investigate and validate the diagnostic power of CT density and other findings of the upper gastrointestinal tract on noncontrast PMCT in the diagnosis of UGIB.

## Materials and methods

This case-control study was approved by the Ethical Committee of the participating institution (Ethical Committee No. 2076, June 9, 2008). The protocol complied with the 1964 Declaration of Helsinki and its later amendments (or with comparable ethical standards). Written informed consent for the use of cadavers in our study was obtained from all families of the deceased participants. The data were accessed between April 2021 and March 2022.

### Participants

Cases of nontraumatic in-hospital deaths at our university hospital between April 2009 and December 2020 were included if the patients were aged 18 years or older at death and had undergone both noncontrast PMCT and conventional autopsy; otherwise, they were excluded. A total of 593 patients satisfied these criteria. All cadavers were stored in the supine position at room temperature from the time of death until the PMCT examination and subsequent autopsy. Using the results of the pathological autopsy as a reference standard, the patients were divided into two groups: 27 with UGIB and 566 without UGIB. We estimated the minimum sample size for representing cases without UGIB to be 83, using the conditions of a 95% confidence level and a 10% margin of error. Using a random number generator, we reduced the number of cases without UGIB to 200 and allocated all cases evenly into the derivation and validation sets. Finally, a derivation set, including 13 patients with UGIB and 100 patients without UGIB, and a validation set, including 14 patients with UGIB and 100 patients without UGIB, were generated. Cases with insufficient upper gastrointestinal content on PMCT were excluded after image analysis. Fig 1 summarizes the patient inclusion process. For patients in the derivation and validation sets, a history of anticoagulation or antiplatelet therapy, the time interval between the last meal (via oral or nasogastric tube) and death, and the time interval between death and PMCT were recorded.

### CT scanning

PMCT was performed using ROBUSTO (Hitachi Medical, Japan) or Aquillion (Toshiba Medical, Japan) multidetector CT devices. The scan parameters were as follows: scanning mode, helical; slice thickness, 2.5 mm; slice interval, 1.25 mm; rotation time, 0.5 s; tube voltage, 120

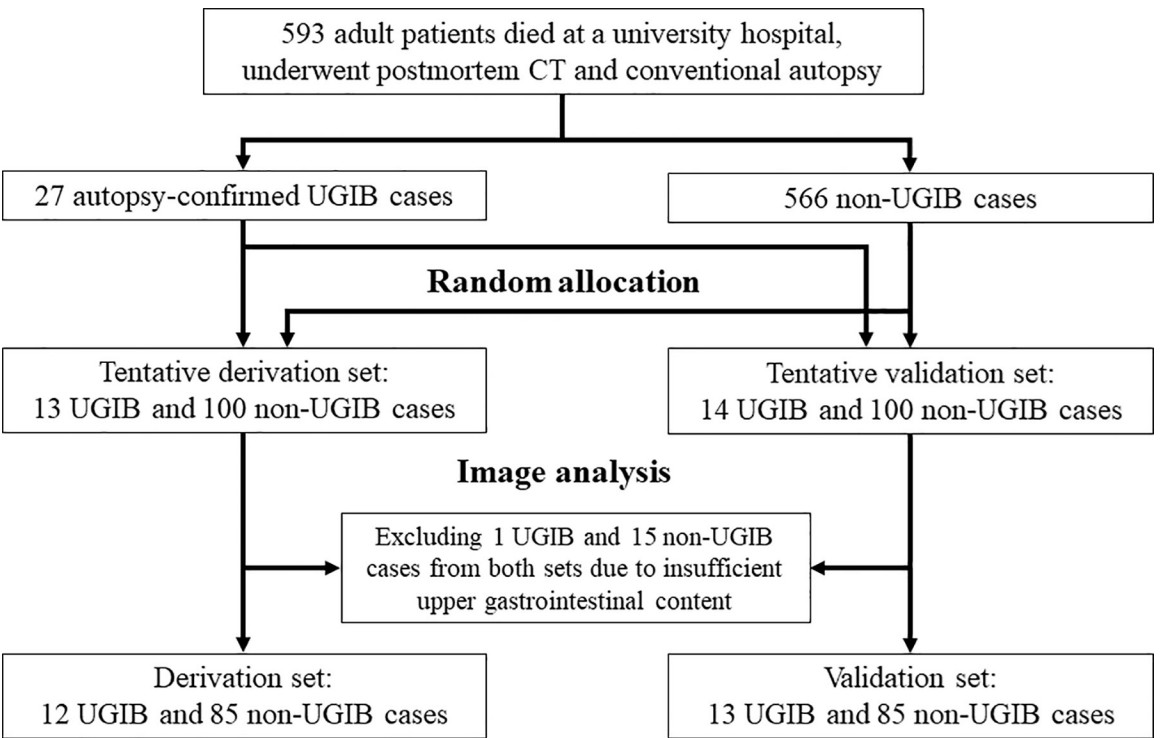

**Fig 1. Flowchart of the case inclusion process for the derivation and validation sets.** UGIB, upper gastrointestinal bleeding.

kVp; tube current, 250 mA. Image reconstruction was performed at a 5-mm thickness with a 350-mm field of view and a 512 × 512 image matrix. No contrast medium was administrated. All data were transferred to an image server in Digital Imaging and Communication in Medicine format.

## Image analysis

Images were analyzed using an open-source medical image viewer, Horos (version 4.0.0, Horos Project, https://horosproject.org/). As for the derivation set, all images were interpreted by a board-certified radiologist (N.O.) with 5 years of experience in PMCT, supervised by another board-certified radiologist (W.G.) with 16 years of experience, who were blinded to the participants' clinical information and autopsy results. The interpreter placed regions of interest approximately 100 mm$^2$ in size on the upper gastrointestinal contents, including the highest CT value area but avoiding streak artifacts, and recorded the mean CT value (Hounsfield Unit, HU; a standardized scale used in CT scanning to measure and compare the radiodensity of various substances: -1000 HU for air, 0 HU for water, and +1000 HU and beyond for dense materials like bone). If the area of the upper gastrointestinal contents was insufficient and a 100-mm$^2$-size region of interest was not drawable at any place, the case was excluded. The optimal CT cutoff value to diagnose UGIB was determined through a receiver operating characteristic (ROC) analysis with the Youden index. To elucidate CT findings characteristic of true-positive and false-positive cases, that is, cases with a higher CT value of upper gastrointestinal content than the optimal CT cutoff value, an additional assessment was conducted. The following findings of the high-density upper gastrointestinal contents were recorded: location (ventral, dorsal, homogeneous, or heterogeneous); expected nature of the high-density contents considering the shape of food residue or feces, the presence/absence of fluid-air level,

and homogeneity/inhomogeneity of the content (liquid or solid); presence of bubbles inside or close ($\leq$10 mm) to the high-density contents (present or absent); the relationship of the bubbles mentioned above to the high-density contents if present (inside or outside); the maximum size of the bubbles mentioned above if present (mm); and presence of CT value >100 HU because the CT value of the hematoma rarely exceed 100 HU (yes or no) [14,15]. Among these findings, candidates for useful imaging findings to differentiate true-positives and false-positives were extracted statistically, as mentioned later.

For the validation set, all images were assessed by another radiologist (M.I.) with 15 years of experience in PMCT, who was blinded to the participant's clinical information and autopsy results. The interpreter measured the CT values of the upper gastrointestinal contents in the same manner as for the derivation set. Patients with insufficient upper gastrointestinal contents were excluded.

## Autopsy technique

Conventional autopsies, including investigations of the upper gastrointestinal tract, were performed by board-certified pathologists immediately after PMCT (within 1 h) in all cases. The pathologists were informed of the patient's clinical histories but were blinded to the radiologists' PMCT reports. Gross inspection and histological analyses were performed for each organ, and the cause of death was determined for each case. UGIB was diagnosed when hemorrhagic contents or hematomas were macroscopically observed in the upper gastrointestinal tract. A minute or microscopic hemorrhage was not recorded as UGIB as it was unlikely to cause death or critical conditions.

## Statistical analysis

Statistical analysis was performed using EZR software (Saitama Medical Center, Jichi Medical University) [16]. Fisher's exact test and the Mann–Whitney U test were used to examine the statistical significance of differences. Statistical significance was set at a $P$ value < 0.05. The ROC curve with the Youden index was used to define the optimal CT cutoff values for the upper gastrointestinal contents. The area under the curve (AUC), sensitivity, and specificity were also calculated [17,18].

# Results

## Subject characteristics in the derivation set

Of the 113 cases, one with UGIB and 15 without UGIB were excluded due to insufficient upper gastrointestinal contents for measurement on the PMCT images. As a result, 12 patients with UGIB and 85 without UGIB comprised the derivation set (97 cases). The causes of death in the derivation set were as follows (number of cases with UGIB vs. without UGIB): respiratory failure (1 vs. 32), liver failure (0 vs. 6), renal failure (0 vs.1), sepsis (1 vs. 6), multiorgan failure (0 vs. 2), heart disease (2 vs. 12), upper gastrointestinal hemorrhage (5 vs. 0), extra-gastrointestinal hemorrhage (1 vs. 2), malignant tumor (1 vs. 11), intracranial lesion (1 vs. 8), peritonitis (0 vs. 2), and unidentified (0 vs.3). The clinical characteristics of the other participants are summarized in Table 1.

## Determining the CT cutoff value using the derivation set

In the derivation set, the mean CT value of the upper gastrointestinal contents was 48.2 HU (standard deviation, 19.3 HU) in cases with UGIB and 22.8 HU (standard deviation, 19.1 HU) in cases without UGIB (Mann–Whitney U test, $p$ < 0.001). The ROC curve for the CT values

**Table 1. Demographic and clinical characteristics of the cases with/without upper gastrointestinal bleeding in the derivation set.**

| | | UGIB | Non-UGIB | *p* value |
|---|---|---|---|---|
| | | (n = 12) | (n = 85) | |
| Gender | Male | 7 | 55 | 0.75[a] |
| | Female | 5 | 30 | |
| Age (y; mean +/- SD) | | 60.0 +/- 16.9 | 66.6 +/- 13.4 | 0.2[b] |
| Anticoagulation/antiplatelet therapy[c] | Yes | 3 | 15 | 0.85[a] |
| | No | 9 | 65 | |
| Time interval between death and PMCT (h; mean +/- SD) | | 8.0 +/- 5.8 | 9.9 +/- 9.9 | 0.55[b] |
| The time interval between the last meal and death[d] | <24 h | 3 | 8 | 0.17[a] |
| | ≥24 h | 9 | 67 | |

Notes

[a], Fisher's exact test

[b], Mann–Whitney U test

[c], medication history was not available for five cases and excluded from analysis

[d], the timing of the last meal was not available for 10 cases and excluded from analysis; PMCT, postmortem computed tomography; SD, standard deviation; UGIB, upper gastrointestinal bleeding.

of the upper gastrointestinal contents is illustrated in Fig 2. According to the ROC curve analysis using the Youden index, the optimal CT cutoff value was 27.7 HU (sensitivity, 91.7%; specificity, 81.2%; AUC, 0.898).

## PMCT findings in cases with high CT value content in the derivation set

A total of 27 cases in the derivation set had upper gastrointestinal contents of ≥27.7 HU. Among them, 11 had UGIB, and 16 did not have UGIB. The results of the additional assessments for these cases are summarized in Table 2.

In cases with UGIB, the high-density contents were more solid, and the size of the bubbles was greater than that in cases without UGIB. All bubbles in cases with UGIB were > 4 mm, while it were < 4 mm in all cases without UGIB. Differences between cases with and without UGIB in other PMCT findings were not significant. After using the CT cutoff value of ≥27.7 HU, the sensitivity and specificity increased if the additional criteria were used as follows: solid nature of the high-density contents among cases with a CT cutoff value ≥27.7 HU, with a sensitivity of 63.6% and specificity of 87.5%, and existence of bubbles ≥4 mm inside or close to the high-density contents among cases with a CT cutoff value ≥27.7 HU and bubbles, with a sensitivity of 100% and specificity of 100%. Additionally, in four of the five patients who died from UGIB, the CT values of the upper gastrointestinal contents were above the optimal cutoff value.

## Subject characteristics of the validation set

In the validation set of 114 cases, one with UGIB and 15 without UGIB were excluded because of insufficient upper gastrointestinal contents. Consequently, 13 patients with UGIB and 85 patients without UGIB (98 cases) were included in the validation set. The causes of death were as follows (number of cases with UGIB vs. without UGIB): respiratory failure (5 vs. 40), liver failure (1 vs. 3), renal failure (0 vs. 1), sepsis (0 vs. 5), multiorgan failure (0 vs. 1), heart disease (1 vs. 4), upper gastrointestinal hemorrhage (5 vs. 0), extra-gastrointestinal hemorrhage (0 vs. 4), malignant tumor (0 vs. 17), intracranial lesion (0 vs. 8), peritonitis (1 vs. 1), and unidentified (0 vs. 1). The clinical characteristics of the other participants are summarized in Table 3.

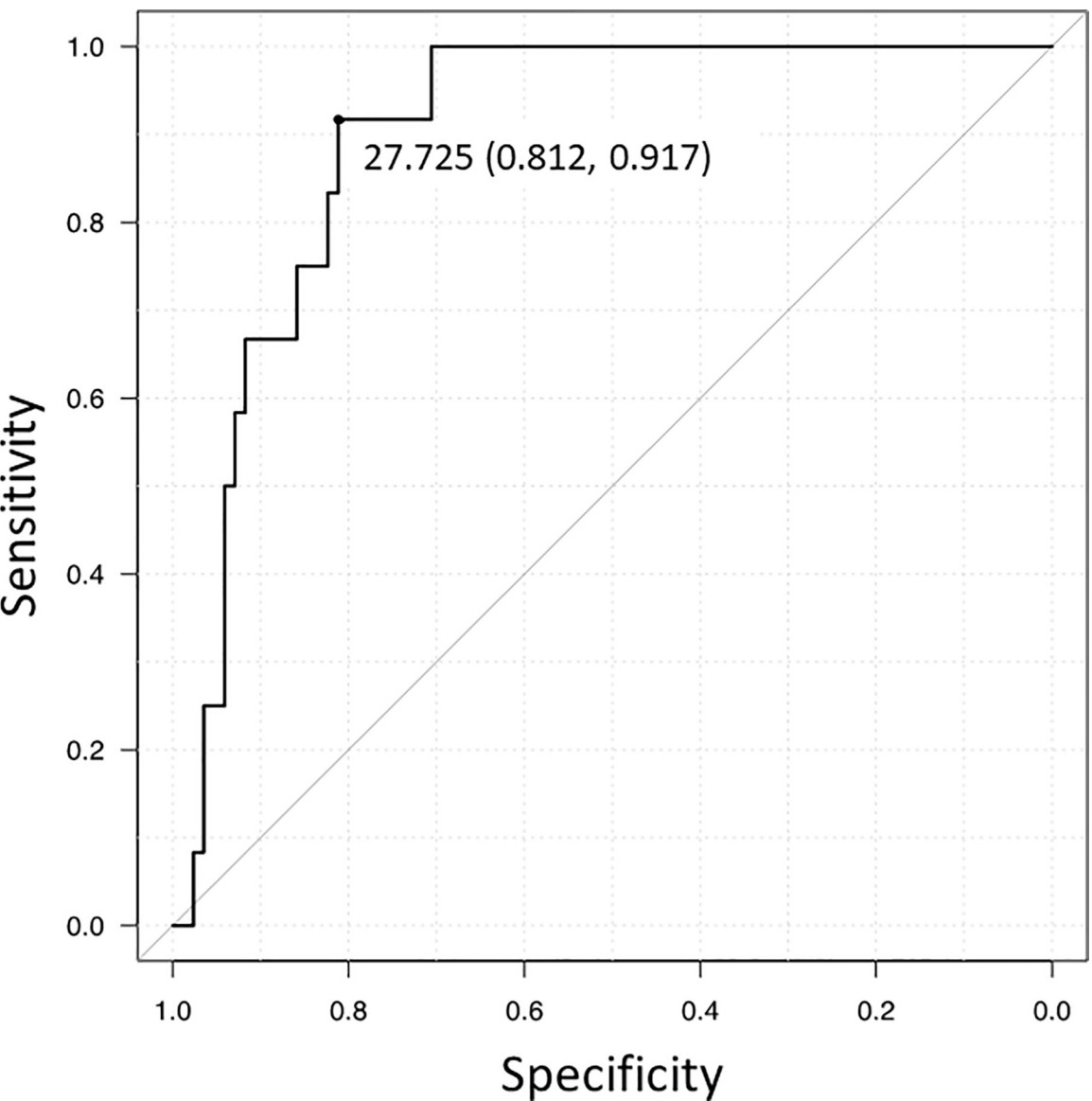

**Fig 2. ROC curve for CT value of the upper gastrointestinal content.** Receiver operating characteristic (ROC) curve for computed tomography (CT) value of the upper gastrointestinal content was depicted to distinguish cases with and without upper gastrointestinal bleeding in the derivation set. The optimal CT cutoff value was 27.7 HU according to the Youden index (sensitivity, 91.7%; specificity, 81.2%; area under the curve, 0.898).

### Testing the CT cutoff value and additional PMCT findings in the validation set

In the validation set, the mean CT value of the upper gastrointestinal contents was 51.8 HU (standard deviation, 21.4 HU) in cases with UGIB and 40.9 HU (standard deviation, 104.7 HU) in cases without UGIB (Mann–Whitney U test, $p < 0.001$). Thirty of 98 cases in the validation set had $\geq$27.7 HU gastrointestinal content, comprising 11 cases with UGIB and 19 without UGIB. In diagnosing UGIB, the sensitivity and specificity for the CT cutoff value $\geq$27.7 HU were 84.6% and 77.6%, respectively. Using the criteria of both the CT cutoff value $\geq$27.7 HU and the solid nature of the high-density contents, overall sensitivity and specificity

**Table 2. Postmortem CT findings among cases with high-density gastrointestinal contents (≥27.7 HU) in derivation set for differentiating true-positive and false-negative cases.**

| | | | UGIB | Non-UGIB | p value |
|---|---|---|---|---|---|
| | | | (n = 11) | (n = 16) | |
| High-density contents | Property | Liquid | 4 | 14 | 0.012[a],* |
| | | Solid | 7 | 2 | |
| | Location | Ventral | 0 | 0 | 0.075[a] |
| | | Dorsal | 8 | 7 | |
| | | Homogenous | 2 | 9 | |
| | | Heterogeneous | 1 | 0 | |
| | CT value | <100 HU | 0 | 2 | 0.5[a] |
| | | ≥100 HU | 11 | 14 | |
| Bubbles inside/close to the high-density content | Present | | 5 | 5 | 0.687[a] |
| | Absent | | 6 | 11 | |
| | Maximum size (mm; mean, [range]) | | 7.6 [4.6–11.6] | 2.9 [1.9–3.9] | 0.008[b],* |
| | Maximum size (n) | ≥4 mm | 5 | 0 | 0.008[a], * |
| | | <4 mm | 0 | 5 | |
| | Relationship to high-density contents | Inside | 3 | 5 | 0.44[a] |
| | | Outside | 2 | 0 | |

Notes

* statistically significant

[a] Fisher's exact test

[b] Mann–Whitney U test; HU, Hounsfield unit; UGIB, upper gastrointestinal bleeding.

in the validation set changed to 61.5% and 96.5%, respectively. Using the criteria of both the CT cutoff value ≥27.7 HU and the existence of bubbles ≥4 mm inside or close to the high-density contents, sensitivity and specificity were 38.5% and 98.8%, respectively. Representative true-positive, false-positive, false-negative, and true-negative cases of UGIB are shown in Fig 3. As a supplement, for all five patients who died from UGIB, the CT values of the upper gastrointestinal contents were above the optimal cutoff value.

**Table 3. Demographic and clinical characteristics in cases with and without upper gastrointestinal bleeding in the validation set.**

| | | UGIB | Non-UGIB | p value |
|---|---|---|---|---|
| | | (n = 13) | (n = 85) | |
| Gender | Male | 7 | 61 | 0.21[a] |
| | Female | 6 | 24 | |
| Age (y; mean +/- SD) | | 66.1 +/- 18.5 | 66.5 +/- 14.3 | 0.78[b] |
| Anticoagulation/antiplatelet therapy[c] | Yes | 4 | 17 | 0.52[a] |
| | No | 9 | 61 | |
| Time interval between death and PMCT (h; mean +/- SD) | | 10.0 +/- 6.9 | 9.7 +/- 9.1 | 0.7[b] |
| The time interval between the last meal and death[d] | <24 hours | 0 | 18 | 0.15[a] |
| | ≥24 hours | 12 | 58 | |

Notes

[a], Fisher's exact test

[b], Mann–Whitney U test

[c], medication history was not available for seven cases and excluded from analysis

[d], the timing of the last meal was not available for 10 cases and excluded from analysis; PMCT, postmortem computed tomography; SD, standard deviation; UGIB, upper gastrointestinal bleeding.

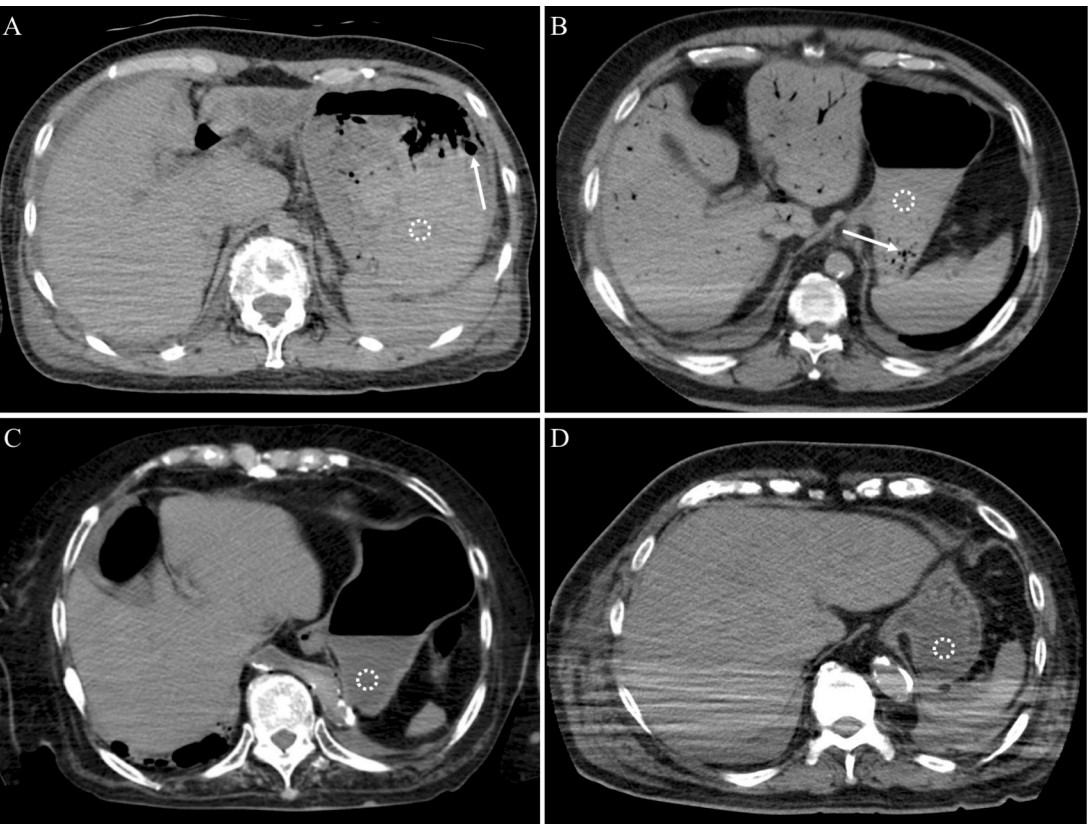

**Fig 3. Representative CT images of true-positive, false-positive, false-negative, and true-negative cases of UGIB.** (A) A noncontrast postmortem CT image of a 65-year-old woman in the validation set who died of ruptured esophageal varices. A solid-natured high-density gastric content (CT value, 53.4 HU) was observed in the stomach adjacent to some bubbles (size, ≥4 mm; arrow), indicating bleeding. A subsequent conventional autopsy revealed that the high-density gastric content was a fresh hematoma (true-positive UGIB case). Dotted circles indicate where the CT value was measured. (B) A noncontrast postmortem CT image of a 74-year-old man in the derivation set who died of respiratory failure. A liquid-natured high-density gastric content (CT value, 63.2 HU) was observed in the stomach, inside of which bubbles (size, <4 mm; arrow) were observed, suggesting bleeding. A subsequent conventional autopsy revealed that the high-density gastric content was food residue (false-positive UGIB case). (C) A noncontrast postmortem CT image of an 84-year-old woman in the validation set who died of heart disease. A low-density content (CT value, 15.7 HU) was noted in the stomach, suggesting food residue. A subsequent conventional autopsy revealed the low-density gastric content to be hematoma (false-negative UGIB case). (D) A noncontrast postmortem CT image of a 79-year-old man in the validation set who died of respiratory failure. A low-density content (CT value, 9.4 HU) was observed in the stomach, suggesting food residue. A subsequent conventional autopsy revealed that the low-density gastric content was a food residue (true-negative UGIB case).

## Discussion

We elucidated that measuring the CT values of the upper gastrointestinal contents on noncontrast PMCT is useful for diagnosing UGIB in deceased patients in this case-control study. The proposed criterion of ≥27.7 HU was robustly validated owing to its high diagnostic power and reproducibility. Previous antemortem CT studies have reported that the CT value of extravascular blood was 20–40 HU in the unclotted type and 40–70 HU in the clotted type [8,14]. Thus, UGIB observed in the present study may be a combination of unclotted and clotted blood samples. An ex vivo study reported that the higher the hematocrit, the higher the density of blood to above 100 HU [19]. While the cutoff CT value proposed in the present study differentiated UGIB and non-UGIB adequately, there were some false-positive cases. These false-positive cases may have high-density gastrointestinal contents, such as food residue and medication, rendering a clear differential diagnosis difficult [11].

In the present study, we explored additional candidate imaging findings to enhance the accuracy in distinguishing true-positive UGIB from false-positive UGIB. One was the solid nature of the high-density contents in the upper gastrointestinal tract, which improved the specificity by 18.9% but decreased the sensitivity by 23.1%. As reported previously, the solidity of the hematoma is associated with a high degree of clot contraction, and blood concentrates (or coagulates) rapidly compared to nonbiological gastrointestinal contents. Therefore, the solid components may distinguish UGIB from other liquid-like gastrointestinal contents.

Another distinguishing candidate imaging finding was the existence of bubbles ≥4 mm inside or close to the high-density contents where measurement was conducted. We first hypothesized that food residues would include small amounts of air and bubbles caused by mastication and swallowing, whereas hematomas would not. However, in the present study, a larger bubble size was more likely to be observed in the UGIB cases than in the non-UGIB cases. The mechanism by which this phenomenon occurs could be explained as follows: the patient would start fasting on developing UGIB. Therefore, hematomas caused by UGIB are fresher than food residues and have less time to release gas. Otherwise, the relatively larger bubbles in the hemorrhage group may have been due to the higher viscosity of the blood clots.

The present study had several limitations. First, the cohort comprised in-hospital death cases from a single institute, and PMCT was performed immediately after death. Therefore, direct application of these results to putrid forensic cases is not possible. Second, as the aim of the study was to identify the presence of UGIB, the cause of death in patients with UGIB might not necessarily be UGIB. Third, a detailed examination was not conducted on the CT values for each subject's oral medication as it was exceedingly challenging to assess the CT values for all oral medications.

In conclusion, in diagnosing UGIB on noncontrast PMCT, a CT cutoff value of ≥27.7 HU was a reproducible diagnostic criterion.

## Supporting information

**S1 File.**
(XLSX)

## Author Contributions

**Conceptualization:** Masanori Ishida, Wataru Gonoi, Osamu Abe.

**Data curation:** Naomasa Okimoto, Masanori Ishida.

**Formal analysis:** Naomasa Okimoto.

**Funding acquisition:** Masanori Ishida, Wataru Gonoi.

**Investigation:** Naomasa Okimoto, Masanori Ishida, Kotaro Fujimoto, Keisuke Nyunoya, Mariko Kurokawa, Go Shirota, Hiroyuki Abe.

**Methodology:** Masanori Ishida, Wataru Gonoi.

**Project administration:** Wataru Gonoi.

**Software:** Naomasa Okimoto.

**Supervision:** Wataru Gonoi, Tetsuo Ushiku, Osamu Abe.

**Writing – original draft:** Naomasa Okimoto.

**Writing – review & editing:** Masanori Ishida, Wataru Gonoi, Kotaro Fujimoto, Keisuke Nyunoya, Mariko Kurokawa, Go Shirota, Hiroyuki Abe.

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
