## [Decision Letter · Decision Letter 0]

17 Mar 2024

PONE-D-24-03102Cutoff CT value can identify upper gastrointestinal bleeding on postmortem CT: development and validation studyPLOS ONE

Dear Dr. Gonoi,

Thank you for submitting your manuscript to PLOS ONE. After careful consideration, we feel that it has merit but does not fully meet PLOS ONE’s publication criteria as it currently stands. Therefore, we invite you to submit a revised version of the manuscript that addresses the points raised during the review process.

We look forward to receiving your revised manuscript.

Kind regards,

Abhishek Das, MD

Academic Editor

PLOS ONE

Journal Requirements:

2. Please expand the acronym “JSPS” (as indicated in your financial disclosure) so that it states the name of your funders in full.

Reviewers' comments:

Reviewer's Responses to Questions

**Comments to the Author**

1. Is the manuscript technically sound, and do the data support the conclusions?

Reviewer #1: Yes

Reviewer #2: Yes

Reviewer #3: Yes

2. Has the statistical analysis been performed appropriately and rigorously? 

Reviewer #1: I Don't Know

Reviewer #2: Yes

Reviewer #3: Yes

3. Have the authors made all data underlying the findings in their manuscript fully available?

Reviewer #1: Yes

Reviewer #2: Yes

Reviewer #3: Yes

4. Is the manuscript presented in an intelligible fashion and written in standard English?

Reviewer #1: Yes

Reviewer #2: Yes

Reviewer #3: Yes

5. Review Comments to the Author

Reviewer #1: Authors are using the terms CT and HU very frequently [for example, “mean CT value of the upper gastrointestinal contents was 48.2 Hounsfield units (HU)”]. Since this paper is going to be read by forensic people, who are not generally trained in radiology, the authors must explain these two concepts in simple terms. Rest of the paper appears okay to me.

Reviewer #2: In total a very informative and practically significant article.

Statistical tests are ok, the charts and the figures are adequate.

Data collection method was ok, however, it is mentioned (line no. 19, 56) that it is retrospective case control study and at line No. 58-60 it is mentioned that “Written informed consent for the use of cadavers in our study was obtained from all families of the deceased participants” needs explanation.

Reviewer #3: The article is a good research work in the field of Virtual Autopsy. This will help in upcoming centers of Virtual Autopsy for a more confirmatory diagnosis of Upper GI bleeding. The cutoff values mentioned will add to Medical literature.

6. PLOS authors have the option to publish the peer review history of their article (what does this mean?). If published, this will include your full peer review and any attached files.

Reviewer #1: No

Reviewer #2: **Yes: **Amar Jyoti Patowary, Professor & Head, Deptt. of Forensic Medicine, NEIGRIHMS, Shillong, India

Reviewer #3: No

---

## [Author Response · Author response to Decision Letter 0]

21 Mar 2024

Response to the Editors and the Reviewers

Thank you very much for your letters of March 8th, 2024, regarding our manuscript entitled, "Cutoff CT value can identify upper gastrointestinal bleeding on postmortem CT: development and validation study" (PONE-D-24-03120). We appreciate the reviewer's and the editor's comments.

Emails received from the editorial staff and reviewers are in blue italics, and our responses are in black after the ">" symbol.

The editorial wrote (Mar/8/2024):

> We have checked our manuscript style. We have added figure title in our main manuscript.

2. Please expand the acronym “JSPS” (as indicated in your financial disclosure) so that it states the name of your funders in full.

> We have spelled out JSPS (Japan Society for the Promotion of Science) in the submission system and in the Cover letter.

> We have uploaded the minimal anonymized data set as Supporting Information files.

> We have checked the references. 

Review Comments to the Author

Reviewer #1: Authors are using the terms CT and HU very frequently [for example, “mean CT value of the upper gastrointestinal contents was 48.2 Hounsfield units (HU)”]. Since this paper is going to be read by forensic people, who are not generally trained in radiology, the authors must explain these two concepts in simple terms. Rest of the paper appears okay to me.

> We have added some introduction for CT and HU as follows: 

“Computed tomography (CT) is a medical imaging technique that uses X-rays and computer processing to create detailed cross-sectional images of the body” (line 38), 

“a standardized scale used in CT scanning to measure and compare the radiodensity of various substances: -1000 HU for air, 0 HU for water, and +1000 HU and beyond for dense materials like bone” (line 99).

Reviewer #2: In total a very informative and practically significant article.

Statistical tests are ok, the charts and the figures are adequate.

Data collection method was ok, however, it is mentioned (line no. 19, 56) that it is retrospective case control study and at line No. 58-60 it is mentioned that “Written informed consent for the use of cadavers in our study was obtained from all families of the deceased participants” needs explanation.

> We prospectively designed an overall outline of the research plan 16 years ago and collected data accordingly. While the current study was part of the original plan, the specific measurement items were ultimately determined retrospectively. As a result, we had initially described it as a retrospective study, but the overall design can be considered prospective. Therefore, we have removed the word "retrospective" in the Abstract, Materials and Methods, and Discussion.

Reviewer #3: The article is a good research work in the field of Virtual Autopsy. This will help in upcoming centers of Virtual Autopsy for a more confirmatory diagnosis of Upper GI bleeding. The cutoff values mentioned will add to Medical literature.

> Thank you very much. We appreciate your comment. 

> In addition, we have corrected some inconsistencies in terminology and errors that I noticed during the review, leaving the changes tracked, but there are no changes to the content itself.

The typing error was amended: “derivation” changed to “validation” (line 172).

Inconsistencies in terminology was corrected in accordance with Materials and Methods: “inside” changed to “inside or close to” (lines 181 and 213).

---

## [Decision Letter · Decision Letter 1]

22 May 2024

Cutoff CT value can identify upper gastrointestinal bleeding on postmortem CT: development and validation study

PONE-D-24-03102R1

Dear Dr. WATARU GONOI,

We’re pleased to inform you that your manuscript has been judged scientifically suitable for publication and will be formally accepted for publication once it meets all outstanding technical requirements.

Kind regards,

Abhishek Das, MD

Academic Editor

PLOS ONE

---

## [Editor Report · Acceptance letter]

30 May 2024

PONE-D-24-03102R1 

PLOS ONE

Dear Dr. Gonoi, 

I'm pleased to inform you that your manuscript has been deemed suitable for publication in PLOS ONE. Congratulations! Your manuscript is now being handed over to our production team.

Kind regards, 

on behalf of

Dr. Abhishek Das 

Academic Editor

PLOS ONE